# Trends of Stomach Cancer in Central Serbia

**DOI:** 10.3390/medicina57070665

**Published:** 2021-06-28

**Authors:** Miodrag M. Stojanovic, Natasa K. Rancic, Marija R. Andjelkovic Apostolovic, Aleksandra M. Ignjatovic, Mirko V. Ilic

**Affiliations:** Institute for Public Health, Faculty of Medicine, University of Nis, 18000 Nis, Serbia; drmstojanovic@gmail.com (M.M.S.); drmari84@gmail.com (M.R.A.A.); drsalea@yahoo.com (A.M.I.); mirkoilic1971@gmail.com (M.V.I.)

**Keywords:** stomach cancer, incidence, mortality, trend

## Abstract

*Background and Objectives*: Somach cancer is the third most common cause of cancer-related deaths worldwide. The objective of the paper was to analyze the incidence and mortality trends of stomach cancer in Central Serbia in the period between 1999–2017. *Materials and Methods:* trends and annual percentage change (APC) of the incidence and mortality rate with corresponding 95% confidence intervals (CI) were calculated by joinpoint regression analyses. The optimal number of Joinpoints was identified using the Monte Carlo permutation method. The trend was considered to be significantly increasing (positive change) or decreasing (negative change) when the *p*-value was below 0.05 (*p* < 0.05). *Results:* the total number of new cases was 16,914 (10,873 males and 6041 females) and the total number of mortality cases was 14,790 (9348 in and 5442 in females). Almost one third (30.8%) of new cases were registered in the 60–69-year age group, and new cases were significantly more frequent in males than in females (30.8% vs. 29.02%, *p* < 0.001). Joinpoint regression analysis showed a significant decrease of incidence trend in females during the 2000–2015 period with APC of −2.13% (95% CI: −3.8 to −0.5, *p* < 0.001). An insignificant decrease in incidence trend was in males with APC of −0.72% (95% CI: −2.3 to 0.9, *p* = 0.30). According to the joinpoint analysis, a significant decrease of mortality trends both in males during 2000–2015 with APC of −2.21% (95% CI: −1.6 to −7.5, *p* ≤ 0.001 and in females, during the same period, with APC of −1.75% (95% CI: −2.9 to −0.6, *p* < 0.001) was registered. From 2015 to 2017, a significant increase of mortality was registered with APC of 44.5% (95% CI: from 24.2 to −68.1, *p* ≤ 0.001) in females and in males with APC of 53.15% (95% CI: 13.5 to −106.6, *p* ≤ 0.001). *Conclusion:* a significant decrease of stomach cancer incidence trend in females and insignificant decrease of incidence trend in males were determined in Central Serbia. Based on presented results, the mortality trend decreased significantly both in males and in females during 2000–2015, and from 2015 to 2017 we recorded a significant increase in mortality in both sexes. We found significantly more new cases in women than in men in the age group of 40–49, and the mortality of stomach cancer was significantly more frequent among females compared to males in the age groups 30–39, as well as in the 50–59 age group. There is a need for improving recording and registration of new cases of stomach cancer, especially in females. Urgent primary and secondary preventive measures are needed—introducing stomach cancer screening and early detection of premalignant changes. Urgent primary and secondary preventive measures are needed.

## 1. Introduction

Stomach cancer is the third cause of cancer-related deaths worldwide and it is one of the deadliest malignant tumors in men [1]. This cancer is most commonly diagnosed in men in five countries in the world (Mongolia, Kyrgyzstan, China, Bhutan and Vietnam) and currently, there is no country where this cancer is most commonly diagnosed in women [2]. Two-thirds of new cases of stomach cancer are registered in developing countries and survival is still low both in developed and in developing parts of the world [3].

According to GLOBOCAN, stomach cancer was the fifth leading cause of illness and the third leading cause of death across the global population in 2018, and more than one million new cases and 783,000 deaths were registered [3]. The incidence rates of stomach cancer vary significantly according to age, gender, socio-economical status and geographical area [2,3]. Most of the patients with stomach cancer are older than 50 years at the time of diagnosis [4], and the incidence is two times higher in men than in women. The incidence of stomach cancer is high in East Asia, Eastern Europe and parts of Central and South America, and low in Southern Asia, North and East Africa, Western and Northern Europe, North America and Australia [3] (Table 1). 

About 90% of all cancers of the stomach are adenocarcinomas, and primary lymphoma accounts only for 5%–10% [4]. According to the anatomical location and different etiology adenocarcinomas are divided into cardia and non-cardia stomach cancers. Chronic gastritis, inflammation of the stomach lining and *Helicobacter pylori* atrophic gastritis are associated with non-cardia cancers while obesity and gastroesophageal reflux disease (GERD) are risk factors for cardia cancers [5]. Non-cardia gastric cancer accounts for the majority of the cases worldwide and this subtype is the predominant in high-risk areas. Cardia cancer is more homogeneously distributed all over the world and its incidence tends to increase [5].

Stomach cancer occurs in nearly 10% of the persons in which two or more relatives from the same family were affect and it is assessed that the risk among relatives is 2–3 fold higher than in persons without familiar background of this malignancy [6].

Stomach cancer was the most common cause of death in men globally until mid-1990s [7] when it was replaced by lung cancer. The incidence of stomach cancer gradually began to decline in the 1950s, first in the USA [8] and then in Japan in the 1960s [9]. The average five-year survival rate is 26% in Europe, 19% in the United Kingdom (UK), 31% in the USA. Women have the highest survival rate in Iceland—42% [7,8]. Overcoming and surviving the disease has increased significantly in the last four decades, owing to stomach cancer screening, which is mandatory in many Asian countries, so it is detected earlier and treated more effectively [9].

The incidence of stomach cancer decreased before the discovery of the bacterium *Helicobacter pylori*, which can be explained by the widespread use of refrigerators for food storage, both in stores and in households after the World War Two. In the USA, about 80% of households already had refrigerators in the early 1950s; in that period, there was no screening for stomach cancer and no other widely used measures for reducing the incidence [10].

The incidence and mortality of stomach cancer show significant variability between regions in the world depending on the population’s eating habits and culture (high salt intake, products containing N-methyl-N-nitro-N-nitrosoguanidine, additives and preservatives, insufficient intake of fresh fruit and vegetables and thus vitamins C, E and beta carotene), as well as the prevalence of other risk factors: smoking, alcohol, radiation, toxic substance exposure, Epstein-Barr virus infection, pernicious anemia, gastroesophageal reflux, blood type A, low socioeconomic status, genetic predisposition [11,12]. Sex and age are significant risk factors, and stomach cancer in developed parts of the world is 2.2 times more common in men than in women [13].

About half of the world’s population is infected with *Helicobacter pylori*, and the majority of colonized individuals develop chronic inflammation without any symptoms. A long-term carriage of *Helicobacter pylori* significantly increases the risk of developing 10% develop peptic ulcer disease (in 10%), gastric adenocarcinoma (3%) and mucosa-associated lymphoid tissue (MALT) lymphoma <0.1% [14]. According to the Food and Drug Agency (FDA) in the United States of America, chronic *Helicobacter pylori* infection is considered a carcinogenic agent significant for stomach cancer development [15]. Chronic *Helicobacter pylori* infection is a significant risk factor and increases the risk for developing stomach cancer within ten years of infection 5.9 times [16]. There are no available data for *Helicobacter pylori* chronic infection prevalence in the Republic of Serbia [17].

The impact of *Helicobacter pylori* infection, diet, food additives and preservatives and low socioeconomic status are particularly significant factors in the areas with a high incidence of stomach cancer, such as Asia and Latin America. According to the results of a meta-analysis that included 76 prospective cohort studies, the association between diet and high alcohol intake for the occurrence of stomach cancer is significant, as is the intake of food with high salt content [18,19].

Alcohol intake increases the risk of developing stomach cancer by 15% and only applies to high daily alcohol intake. A 30-year study conducted in Lithuania confirmed alcohol intake association with an increased risk of stomach cancer [20]. Results of one meta-analysis showed that even one drink a day was associated with an increased risk of stomach cancers. 

In 2018, the standardized incidence rate for stomach cancer in men was 11.1%/100,000 men and in women 5.4%/100,000 women in Central Serbia. In the structure of all types of cancers, stomach cancer with a share of 3.4% was in the 5th place in men and the 9th place in women with a share of 2.3%. The standardized mortality rate was 7.40/100,000 men and 3.4/100,000 women. As a cause of death from all malignant tumors, in 2018 stomach cancer was in the 4th place with 4.6% in men and the 6th place in women with a share of 4.1% [21].

The objective of the study was to analyze the trend of incidence and mortality of stomach cancer in the population of Central Serbia in the period from 2000 to 2017.

## 2. Material and Method

A descriptive study was applied. The data on the incidence and mortality of stomach cancer in 18 years were retrospectively analyzed and obtained from the population cancer registry of Serbia [21].

Non-standardized, specific and standardized incidence and mortality rates per 100,000 people were calculated. We performed the analysis only for central Serbia without the northern and southern provinces. The data from Kosovo and Metohija have not been available in the population register since 1997. Rates were standardized by the Sagy [22] direct method, and the world population was used as the standard. The data on the population of Serbia were obtained from the 1999, 2002 and 2011 Censuses.

The code for stomach cancer—C16—was used according to the 10th revision of the International Classification of Diseases [23]. Since we used the data from already published official publications, Cancer Incidence and Mortality in Central Serbia—http://www.batut.org.rs/index.php?content=185 accessed on 25 June 2021 and Malignant tumors—http://www.batut.org.rs/index.php?content=2096 accessed on 25 June 2021, from the website of the Institute of Public Health of Serbia.

As we used published official data, the permission of the Ethics Committee was not required.

### Statistical Analysis

Crude rates, age-specific and age-standardized rates (ASRs) of incidence and mortality were calculated per 100,000 inhabitants. The method of direct standardization was performed, and the World population was used as a standard. The data about the population of Central Serbia were obtained from Censuses 1991, 2002, 2011. 

Trends and annual percentage change (APC) of the incidence and mortality rate with corresponding 95% confidence intervals (CI) were calculated by Joinpoint regression analyses. The optimal number of Joinpoints was identified using the Monte Carlo permutation method. For the regression analyses, the Joinpoint Regression Program version 4.8.0.0. was used (available at http://surveillance.cancer.gov/joinpoint accessed on 25 June 2021 ). The trend was considered to be significantly increasing (positive change) or decreasing (negative change) when the *p*-value was below 0.05 (*p* < 0.05).

## 3. Results

From 2000 to 2017, the total number of new stomach cancer cases was 16,914. It was diagnosed in 10,873 (64%) males and 6041 (36%) females (male to female incidence ratio was: 1.8:1). The total number of mortality cases was 14,790 (9348 males and 5442 females), deaths from stomach cancer with a male to female mortality ratio of 1.7:1. We registered 890 new cases and 778 fatal cases of stomach cancer in average in Central Serbia.

Table 2 shows the number of new cases of stomach cancer, crude and age-standardized incidence and mortality rates (per 100,000 population) in the male and female populations of Central Serbia for the period 2000–2017.

In males, the number of new stomach cancer cases ranged from 512 (in 2015) to 740 (in 2017), and the average number of new cases was 604 while the average ASR was 12.8. In females, the incidence of stomach cancer cases ranged from 255 to 415 with an average of 336 new cases. The ARSs in females ranged from 4.2 (in 2013) to 10.6 (in 2016). 

The average ASR mortality rates of stomach cancer during the study period in males was 10.9 and 5.2 in females. In the period 2000–2017, annual ASR of mortality for males ranged from 8.0 (2015) to 16.8 (2016) and for females this rate ranged from 3.5 (2015) to 5.9 (2003).

Table 3 shows the distribution of stomach cancer incidence by sex and age in Central Serbia from 2000 to 2017. 

According to the data in Table 3, stomach cancer was registered both in men and in women under 20 years of age. Almost one third (30.8%) of new cases were recorded in the 60–69-year age group. In this age group, new cases were significantly more frequent in males compared to females (30.8% vs. 29.02%, *p* < 0.001). Of all patients, 27.6% were aged over 70 years at the time of diagnosis. In the 40–49-year age group, new stomach cancer cases were significantly more frequent among females compared to males (17.13% vs. 14.30%, *p* < 0.001). 

Figure 1 shows the incidence trend, based on age-standardized incidence rates in males and females, in the period 2000–2017, in Central Serbia.

Joinpoint regression analysis showed a statistically significant decrease of ASR for stomach cancer incidence in females during the 2000–2015 period with APC of −2.13% (95% CI: form −3.8 to −0.5, *p* < 0.001). In addition, a non-significant decrease of stomach cancer was present in males during the examined study period with APC of −0.72% (95% CI: from −2.3 to −0.9, *p* = 0.30).

The highest number of deaths in both sex groups, over 50%, was in the age group of 70–79 years (Table 4). In the 30–39-year age group (1.28% vs. 1.69%, *p* = 0.045), as well as in the 50–59 age group (9.64% vs. 11.03%, *p* < 0.001), the mortality cases of stomach cancer were significantly more frequent among females compared to males. Conversely, in the 70–79 age group, the mortality cases of stomach cancer were significantly more frequent in males compared to females (29.50% vs. 29.02%, *p* = 0.007).

Figure 2 shows stomach cancer mortality in males and females in the period 2000–2017, in Central Serbia.

Joinpoint regression analysis showed a significant decrease in stomach cancer mortality in males during 2000–2015 with APC of −2.21% annually (95% CI: from −1.6 to −7.5, *p* ≤ 0.001). After that, from 2015 to 2017 significant increase in stomach cancer mortality was registered with APC of 44.5% (95% CI: from 24.2 to −68.1, *p ≤* 0.001).

The situation is the same in females, a significant decrease in stomach cancer mortality during the 2000–2015 period with APC of −1.75% annually (95% CI: from −2.9 to −0.6, *p* < 0.001), and after that in the period 2015–2017 significant increase of stomach cancer mortality happened with APC of 53.15% (95% CI: 13.5–106.6, *p* ≤ 0.001).

## 4. Discussion

Cancers are in the second place as cause of morbidity and mortality in Serbia for many decades and have a great impact on the population, especially as cause of premature death [24].

We determined a decline of the incidence trend of stomach cancer in both men and women from 2000 to 2015 and a significant decline of the incidence trend was found only in women in Central Serbia. We also found a sudden jump in incidence in women from 2015 to 2017. At present, it is unclear whether the rising incidence in females is due to better recording and registration in mentioned period or because of insufficient registration of stomach cancer in several past years.

According to our results there was nearly two times more new registered cases in men than women and both new and fatal cases were registered in all age groups. In the age group of 40–49, there were significantly more new cases in women than in men, while it was the other way round in the age group of 60–69. Over the 60 years of age there were significantly more new cases in men than in women.

Our results showed a significant decrease of stomach cancer mortality trend in Central Serbia, both in males and in females during 2000–2015 and from 2015 to 2017 we recorded a significant increase in mortality in both sexes. The highest number of deaths in both sexes, was in the oldest age group of 70–79 year. In the younger age groups, 30–39-year as well as in the 50–59 age group, the mortality of stomach cancer was significantly more frequent among females compared to males. In the 70–79 age group, the mortality was significantly more frequent in males than in females.

Mortality in younger women could be a result of increased incidence due to higher number of women with genetic predisposition or high prevalence of risk factors, such as smoking, alcohol drinking or undiagnostic of malignancy in earlier phase. Increased mortality in persons over 50 years is not unusual, because both incidence and mortality of many cancers increase in middle aged population and increase with age.

Ilić et al. (2016) found that in the period from 1991 to 2015 there was an increase in the mortality trend in men for all malignant tumors in Central Serbia, except stomach cancer [25]. According to findings of these authors the reasons for this trend have not been completely clear, but it could be related to the stabilization of the political, economic and social situation, and with the introduction of modern diagnostics and the introduction of more efficient treatment methods.

Mihajlović et al. (2013) found that in the period 1999–2009 there was an increase of incidence and mortality trend for all malignant tumors in both sexes in Serbia [26]. According to Serbian Office of Statistics (2000) stomach cancer, lung cancer and colorectal cancer were among ten leading causes of deaths in men, whereas stomach cancer, breast cancer, colorectal cancer, lung cancer and cervical cancer were among twelve leading causes of death in women [27].

Based on our results stomach cancer was among ten leading causes of deaths both in men and women in the population of Central Serbia in the period 2000–2017.

Mihajlovic et al. emphasized that smocking and alcohol consumption were very prevalent in population of Serbia in that period. More than a half of all men and one third of all women actively smoke, while 40% of the total population consume alcohol occasionally or on everyday basis [26].

Some action in controlling risk factors in the population have been introduced by the Serbian health care system, as for example the creation of a legislation framework for tobacco control [27], more effort towards risk-factor reductions is needed [28].

There are data in the literature that indicate a decrease in the incidence and mortality of stomach cancer in both sexes [1,2,3,5]. Stomach cancer incidence is declining both in men and women, significantly more in women since the mid-1980s [26] and a decreasing trend were recorded in in the USA, [6,8], in Sweden, [29], in the Netherlands [30]. In Spain, decrease of mortality from stomach cancer began at the end of the 1960 s and has decreased in recent decades, particularly in Andalusia [31].

In the UK, in the period from 2007 to 2017, stomach cancer was in 17th place regarding incidence [32] while in Central Serbia, in a similar period, stomach cancer was in 5th place in men and in 9th place in women [21]. The incidence ratio of new patients was similar in our study to data from the UK, but stomach cancer in Central Serbia is among the top 10 cancers that cause illness and death, which is not the case in the UK. Both men and women in Central Serbia were younger at the time of stomach cancer diagnosis than those in the UK. This difference can be explained by differences in citizens’ socioeconomic status and prevalence of other risk factors.

Mandatory screening and early detection of the disease in many Asian countries, which had the highest incidence and mortality, led to a decline in mortality. Incidence reduction can not be achieved only by screening and more effective treatment methods [2,3]. In Japan, in addition to mandatory screening for stomach cancer, a change in the population’s eating habits has been observed, including primarily increased intake of fresh, unprocessed fruit and vegetables, which can be considered primary prophylaxis [9].

In a study of Radovanovic Spurnic et al. (2021) the prevalence of *Helicobacter*
*pylori* infection in Central Serbia was significantly higher in HIV-negative controls than in in people living with HIV (50.2% vs. 28.1%). The prevalence of *Helicobacter*
*pylori* infection in Central Serbia is still unknown [17].

In Romania, there is a continuous decline in stomach cancer mortality both in men and in women [33,34]. Although more than 63% [33] of the population in Romania has *Helicobacter pylori* infection, only 0.5% of patients developed stomach cancer, which confirms that there are other significant risk factors that contribute to the development of stomach cancer [14,33].

In literature, what is mentioned as significant for the prevention of stomach cancer are lifestyle changes, smoking cessation, alcohol intake cessation, change in eating habits, reduced intake of canned and processed foods and increased intake of fresh fruits and vegetables, treatment of *Helicobacter pylori* infection, stomach cancer screening, monitoring of premalignant changes [2,34,35].

## 5. Conclusions

We determined a decline of stomach cancer incidence in Central Serbia, in both men and women from 2000 to 2015. The significant decline of this trend was determined only in women. We also found a sudden jump in incidence only in women from 2015 to 2017. Men were approximately twice as likely to develop this disease as women, but among patients under the age of 60, there were significantly more women than men. A substantially greater number of young women, from age-groups 30 to 39 and 50 to 59, died from stomach cancer than men. The decline of the stomach cancer mortality trend was significant for both men and women in the period 2000–2015 but in the period 2015–2017 we recorded a significant increase of mortality in both sexes. There is a need to improve the recording and registration of new cases of stomach cancer, especially in females. Urgent primary and secondary preventive measures are needed, introducing of stomach cancer screening and early detection of premalignant changes.

## Figures and Tables

**Figure 1 medicina-57-00665-f001:**
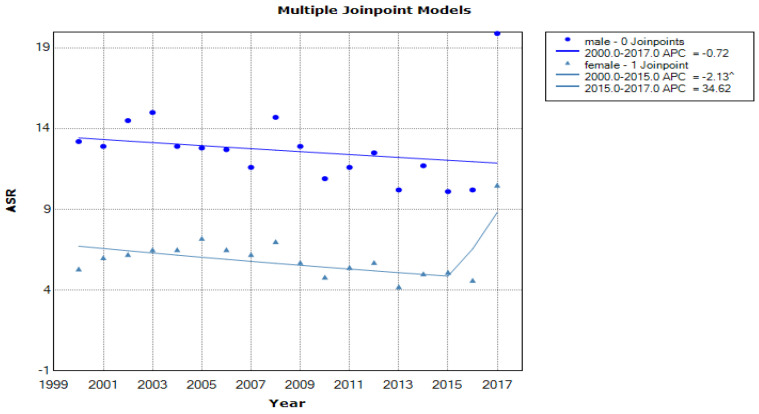
Incidence trend based on age-standardized incidence rates in stomach cancer in Central Serbia according to sex in the period from 2000 to 2017.

**Figure 2 medicina-57-00665-f002:**
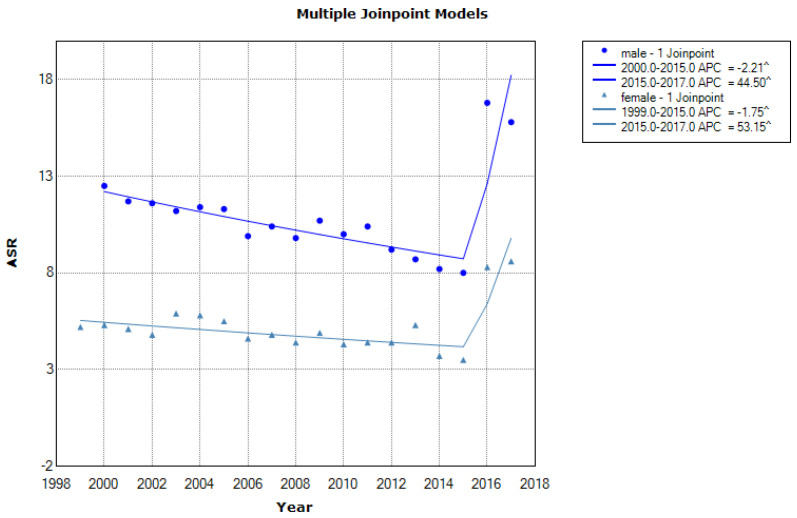
Mortality trend based on age-standardized mortality rates in stomach cancer in Central Serbia according to sex in the period from 2000–2017 with annual percentage change (APC).

**Table 1 medicina-57-00665-t001:** Countries with the highest and lowest incidence and mortality rates of stomach cancer in 2018.

Country	Incidence		Country	Mortality	
	Males	Females		Males	Females
South Korea	57.8	23.8	Mongolia	36.7	15.6
Mongolia	47.2	23.5	Kyrgyzstan	26.7	9.1
Japan	40.7	16.0	China	25.0	10.4
China	29.5	12.3	Vietnam	19.7	8.7
Chile	26.9	10.3	Chile	17.9	6.4
The UK	5.3	2.6	Australia	3.1	1.5
Canada	2.9	5.2	Canada	2.8	1.7
Philippines	4.8	3.0	Sweden	2.7	1.6
Sweden	4.2	2.4	Egypt	7.1	2.5
Egypt	3.1	2.7	The USA	2.3	1.7

Source: https://gco.iarc.fr/today (accessed 9 October 2018).

**Table 2 medicina-57-00665-t002:** The number of new cases, crude and age-standardized incidence and mortality rates (per 100,000 population) of stomach cancer in male and female population of Central Serbia, 2000–2017.

	Incidence						Mortality					
	Males			Females			Males			Females		
Year	New cases	CR	ASR	New cases	CR	ASR	Fatal cases	CR	ASR	New cases	CR	ASR
2000	599	22.3	13.2	290	10.3	5.3	591	22.1	12.5	313	11.1	5.3
2001	595	22.3	12.9	328	11.7	6.0	557	20.9	11.7	337	10.5	5.1
2002	667	25	14.5	344	12.2	6.2	577	21.7	11.6	300	10.7	4.8
2003	705	26.5	15.0	373	13.3	6.5	556	21.0	11.2	303	11.4	5.9
2004	617	23.3	12.9	383	13.7	6.5	566	21.4	11.4	353	12.6	5.8
2005	616	23.3	12.8	415	14.9	7.2	559	21.2	11.3	335	12.0	5.5
2006	609	23.1	12.7	359	12.9	6.5	487	18.5	9.9	283	10.2	4.6
2007	542	20.5	11.6	348	12.5	6.2	515	19.6	10.4	300	10.8	4.8
2008	698	26.6	14.7	399	14.4	7	506	19.4	9.8	283	10.3	4.4
2009	609	23.3	12.9	329	11.9	5.7	312	21.1	10.7	311	11.3	4.9
2010	536	20.6	10.9	284	10.3	4.8	517	19.9	10	279	10.2	4.3
2011	567	21.9	11.6	304	11.1	5.4	533	20.6	10.4	276	10.1	4.4
2012	615	23.8	12.5	336	12.3	5.7	497	19.3	9.2	311	11.5	4.4
2013	517	20.1	10.2	255	9.4	4.2	457	18.1	8.7	325	12.1	5.3
2014	591	22.1	11.7	314	11.7	5	460	18.1	8.2	248	9.2	3.7
2015	512	20.1	10.1	307	11.4	5.1	435	17.2	8.0	229	8.6	3.5
2016	538	21.2	10.2	273	21.5	10.6	632	18.4	16.8	320	11.0	4.6
2017	747	21.8	10.4	467	13.0	5.7	591	17.3	15.8	336	9.3	8.6

CR–crude rate, ASR–age-standardized rate.

**Table 3 medicina-57-00665-t003:** The distribution of stomach cancer incidence by gender and age groups.

Age-Group	Total		Male		Female		*p*-Value
	*n*	%	*n*	%	*n*	%	
0–19	22	0.10	12	0.11	10	0.17	0.340
20–29	160	0.90	104	0.96	56	0.93	0.847
30–39	810	4.80	533	4.90	277	4.59	0.352
40–49	2422	14.30	1387	12.76	1035	17.13	<0.001
50–59	3635	21.50	2371	21.82	1264	20.92	0.171
60–69	5200	30.80	3447	31.72	1753	29.02	<0.001
70–79	4660	27.60	3014	27.73	1646	27.25	0.434
**∑**	16,909		10,868		6041		

**Table 4 medicina-57-00665-t004:** Distribution of stomach cancer mortality by gender and by age groups.

Age-Group	Total		Male		Female		*p*-Value
	*n*	%	*n*	%	*n*	%	
0–19	5	0.03	4	0.04	1	0.02	0.436
20–29	42	0.28	22	0.24	20	0.37	0.145
30–39	212	1.43	120	1.28	92	1.69	0.045
40–49	916	6.19	577	6.17	339	6.23	0.889
50–59	1501	10.15	901	9.64	600	11.03	0.007
60–69	4250	28.74	2758	29.50	1492	27.42	0.007
70–79	7864	53.17	4966	53.12	2898	53.25	0.891
**∑**	14,790	100.00%	9348	100.00%	5442	100.00%	

## Data Availability

Cancer Incidence and Mortality in Central Serbia—http://www.batut.org.rs/index.php?content=185 and Malignant tumors—http://www.batut.org.rs/index.php?content=2096 (accessed on 25 June 2021).

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
