# Peer review of "Trends of Stomach Cancer in Central Serbia"

_medicina, 2021, doi:10.3390/medicina57070665_

Round 1

Reviewer 1 Report

Dear authors and editor,

The manuscript titled ‘’ Trends of Stomach Cancer in Central Serbia‘’ analyze the trend of incidence and mortality of gastric cancer in the population of central Serbia in the period from 2000 to 2017. I agree with the authors that both mortality and incidence trend of GC are gradually declining worldwide, but the survival rate remains low. Undoubtedly, GC poses a significant challenge for physicians.

The paper is well-organised, the language is correct and the content is understandable. Statistical tests mostly correctly selected and performed. Literature properly selected and up to date. I believe they add some contribution to the literature.

However the manuscript is good, I have some comments that should be clarified.

  1. ,,…..The highest incidence rates in 2018 were registered in South Korea (57.8 vs 23.8), Mongolia (47.2 vs 23.5), Japan (40.7 vs 35 16.0), China (29.5 vs 12.3) and Chile (26.9 vs 10.3). The lowest incidence rates in 2018 were the United Kingdam UK (5.3 vs 2.6), Canada 85.2 vs 2.5), Philippines (4.8 vs 3.0), Sweden (4.2. vs 2.4), Egypt (3.1. vs 2.7). In the parts of the world with high incidence rates, mortality rates are high, too [3] and the highest mortality rates in 2018 were recorded in Mongolia (36.7 vs 15.6), Kyrgyzstan (26.6 vs 9.1), China (25.0 vs 10.4) and Vietnam (19.7 vs 8.7)…..’’- A confusing picture, I suggest presenting it in the form of a table or a figure.
  2. ,,….About 95% of all cancers of the stomach are adenocarcinomas and 5% is primary gastric lymphoma. ….’- Please check the literature. Certainly lymphoma is not 5%. Up to 5-10% of non-adenocarcinomas are GISTs, NETs ...... Lymphoma.
  3. ,,… Adenocarcinomas are divided into cardia and non-cardia cancer based on different etiology….’-Based on the localization, we divide stomach cancer into cardia and non-cardia. However, both types appear to have a different etiology.
  4. ,,…Every fifth new patient was between 20 and 49 years old. In the age group of 40–49, 203 there were significantly more new cases in women than men…’;- how can this be explained?

(as is known, the EOGCs (<45) is likely to have a genetic background and the environmental impact is less significant.). Perhaps it would be worthwhile to analyze this group of patients more carefully?

  1. How common is gastric stump cancer in Serbia?

In conclusion, I support publication of the presented article only after thorough correction.

Thank you for your choice me as a reviewer.

Author Response

Thank You for the sugestions.

I shortened the Introduction and discussion. I added 4 new references from Serbian authors and institution (there are 7 references from Serbian authors or institutions). I also removed the next sentence:  Every fifth new case occurred in people from 20 to 49 years of age. I explain that in discussion.

Reviewer 2 Report

Manuscript ID: medicina-1227890

Title

Trends of Stomach Cancer in Central Serbia

Authors

Miodrag M Stojanovic, Natasa K Rancic, Marija R Andjelkovic Apostolovic, Aleksandra M Ignjatovic, Mirko V Ilic

___________________________________

This is a potentially interesting manuscript addressing the incidence trends of Stomach Cancer in Central Serbia (the authors have to report the overall population covered by the cancer registry).

Frankly, the manuscript is too long (particularly the introduction & Discussion), and the discussion does not focus on the most interesting results.

An “epidemic” rising of GC-related mortality is documented from 2015 to 2017 for both males and females (figure 2); the incidence trend (figure 1) however - shows a significant increase of incidence only in females. The authors' Discussion should specifically /extensively address this point.

This astonishing trend seems more likely to result from the registration procedures than from a real clinical-biological situation. GC-related mortality presumably follows 2-5 years the (subclinical) onset of the more advanced precancerous lesions...which means that something should have happened roughly around 2012 (only affecting women)

The present discussion mostly addresses the world’s trends of cancer-site (cardia versus non-cardia), the GC etiology (H.p), the GC incidence, and the GC mortality... all interesting issues that could be appropriate for a review article... but the readers' expectations are  related to the results as reported by the Authors

Author Response

Thank You for Your suggestions.

  1. I made a table and all statistical data from the Introduction are in Table 1.
  2. I corect data about prevalence of stomach lymphoma to 5-10% and I added a reference number 4.
  3. I added data about non-cardia and cardia adenocarcinoms.
  4. I removed from the Results, Disussion and abstract the following sentence: Every fifth new case occurred in people from 20 to 49 years of age. I explained that in the Discussion.
  5. I added 4 references more about prevalence and mortality of stomach cancer in Serbia.

Round 2

Reviewer 2 Report

MANUSCRIPT TITLE:

Trends of Stomach Cancer in Central Serbia

Authors:

Miodrag M Stojanovic , Natasa K Rancic * , Marija R Andjelkovic Apostolovic, Aleksandra M Ignjatovic , Mirko V Ilic

__________

GENERAL COMMENTS ON THE R1 VERSION.

There are still two major points to be addressed and both of them strongly affect the suitability of the manuscript for publication:

1) Within Discussion, the Authors have to implement a plausible interpretation of a major finding of their study “We also found a sudden jump in incidence only in women from 2015 to 2017.”

In the era of H. pylori eradication, the results (mostly those related to the GC incidence) are really bewildering.

2) If the achieved results prompt the need to “improving recording and registration of new cases of stomach cancer”,  the authors should specifically discuss the points of weakness in the current procedure of cancer registration (possibly not only related to gastric malignancies). Focusing on such a point would demonstrate a trend towards improvement… The weakness in the oncological registration is not a Serbian-restricted problem and any critical approach to the current situation is part of the scientific approach.

Both the above-mentioned issues have to be specifically addressed in a new version of the manuscript.

As stated in my previous comments, it is my personal opinion that the manuscript could be sensibly shortened and the English editing significantly improved.

Author Response

Very Respected Rewiever,

Thank You for Your suggestions.

We improved Introduction, Dissusion and Conlusion.

We tried to explain the sudden jumpr.

Also, we answer on the question about Helicobactre pylori.

We explaned which part of reporting of cancers is have to improved. Methodology of Cancer Register is exalent. Contol of cancer reporting is missing. Reporting is insufficient and reports of all cancers are incomplete. That could be an answer of our results.